# Current Situation of Diagnosis and Treatment of HER2-Positive Metastatic Breast Cancer Patients in China: A Nationwide Cross-Sectional Survey of Doctors

**DOI:** 10.3390/jpm13020365

**Published:** 2023-02-19

**Authors:** Kuikui Jiang, Danyang Zhou, Ruoxi Hong, Qianyi Lu, Fei Xu, Wen Xia, Qiufan Zheng, Shusen Wang

**Affiliations:** 1State Key Laboratory of Oncology in South China, Department of Medical Oncology, Sun Yat-sen University Cancer Center, Guangzhou 510060, China; 2Department of Oncology, Peking University Shenzhen Hospital, Shenzhen 518036, China

**Keywords:** breast cancer, metastasis, target therapy, cancer management

## Abstract

Background: The Advanced Breast Cancer Alliance conducted a nationwide investigation to understand the current situation of the diagnosis and treatment of human epidermal growth factor receptor 2 (HER2)-positive metastatic breast cancer (MBC) patients. Methods: In 2019, electronic questionnaires including basic information about respondents, characteristics of patients, and the present status of diagnosis and treatment were sent to 495 doctors from 203 medical centers covering 28 provinces. Results: The factors that influenced treatment plans included the disease process, the performance status, and the economic status of patients. Regimens and response to neoadjuvant/adjuvant chemotherapy were important factors in the decision of the first-line treatment. Overall, 54% of doctors retained trastuzumab and replaced chemotherapy drugs in second-line treatment regimens for patients with progression-free survival (PFS) ≥ 6 months in the first-line setting, while 52% of participants chose pyrotinib plus capecitabine for patients with PFS < 6 months. Economic factors played an important role in doctors’ decision-making and the varying treatment options for respondents in first-tier, second-tier, and other cities. Conclusions: This large-scale survey regarding the diagnosis and treatment of HER2-positive MBC patients revealed that clinical decisions made by Chinese doctors followed the guidelines, but their choices were constrained by economic factors.

## 1. Introduction

Breast cancer is the most common malignant tumor that affects the health of women worldwide, and its morbidity and mortality ranking is first among female malignancies [1]. The incidence of breast cancer in China has been increasing in recent years, with about 279,000 new cases each year [2]. Approximately 15–20% of patients with breast cancer have an amplification of the human epidermal growth factor receptor 2 (HER2) expression [3]. Previous studies have suggested that detection or early screening of breast cancer with appropriate diagnosis and treatment could significantly improve the long-term survival rates of breast cancer patients and reduce treatment expenses [4]. Nevertheless, about 30% of breast cancer patients will eventually develop distant metastatic disease [5]. HER2-positive breast cancer is highly invasive with a low survival rate, which is an independent prognostic factor for breast cancer [6]. The diagnosis and treatment strategy for these patients has been a topic of concern among doctors and there are many unresolved problems that deserve further investigation and discussion.

The treatment of HER2-positive metastatic breast cancer (MBC) is based on anti-HER2 targeted therapy. Currently, the anti-HER2 targeted drugs are becoming more abundant, including HER2 antibodies (Trastuzumab, Pertuzumab), Tyrosine kinase inhibitors (Lapatinib, Pyrotinib, Neratinib, Tucatinib), antibody-drug conjugates (Trastuzumab-emtansine (T-DM1), DS-8201), novel HER2 antibody (Margetuximab), among others. These drugs have greatly improved the effectiveness of the treatment for HER2-positive advanced breast cancer patients, which prolonged progression-free survival (PFS) and overall survival (OS). However, these drugs are relatively expensive, especially when they have not been entered into the health insurance directory. Some of these drugs are not yet available in China, making it more expensive to buy them abroad. Economic factors play a vital role in the management of HER2-positive MBC patients in China. Economic toxicity has received increasing attention in recent years. For cancer patients, this relates to the economic burden and distress caused by receiving anti-tumor treatment. This may lead to a reduction in the patients’ quality of life (QOL), delay treatment, a reduction of the quality of treatment, and even result in a poor prognosis [7,8,9,10].

First-tier cities refer to the larger cities in China that hold a leading position economically, politically, and in health care. These cities have a certain leading and driving effect compared to other cities. The leading effect of these cities originate from many aspects, and the economic factors are the most important [11]. With the economic and social development, the medical level in China has gradually improved and Chinese doctors are making more contributions to international guidelines. However, China is still a developing country with disproportional social and economic development in different regions. The management of HER2-positive MBC patients requires considerations of not only medical factors, but also social and economic factors, including treatment costs, healthcare insurance, and drug accessibility. At present, there is no large-scale survey based on the focus of Chinese doctors on this issue. Therefore, we conducted this survey to determine the reality surrounding the situation of the diagnosis and treatment of HER2-positive MBC patients in China; to provide a reference for consensuses and guidelines on the management of HER2-positive MBC; and to promote the implementation of standards for the training of breast specialists.

## 2. Materials and Methods

### 2.1. Study Design and Population

This survey was divided into two phases. The first phase was from 30 June to 16 July 2019, while the second phase was from 16 September to 24 October 2019. The population consisted of 72 doctors from 38 hospitals in the Advanced Breast Cancer Alliance (the ABC Alliance) while the second phase included 423 doctors from 203 hospitals nationwide. This survey investigated the basic information of doctors and hospitals, the clinical characteristics of patients, and the present situation of diagnosis and treatment.

### 2.2. Data Collection

Data were collected using a pre-designed, unbiased, and non-leading questionnaire. The electronic questionnaire consisted of 57 questions and was sent to 503 doctors in 203 centers with 495 doctors (98.4%) answering the questions. Their responses were collected and analyzed. Among these questions, there are 4 ranking questions requiring selection of 5 options, which are rated as 100, 80, 60, 40, and 20 scores, and 2 ranking questions requiring the choice of 4 options, which are rated as 100, 70, 40, 10 scores, and 0 score for those are not selected.

### 2.3. Statistical Analysis

Progression-free survival was defined as the time from treatment initiation to first documented disease progression or death from any cause. Overall survival was defined as the time from treatment initiation to death from any cause. The statistical method for ranking questions was the weighted average method. For choice questions, the percentage was calculated by the sample size of selecting this option divided by the total sample size of answering this question. The statistical significance was set at *p* ≤ 0.05 and all tests were two-tailed. SPSS version 24.0 (IBM, New York, NY, USA) was used for statistical analyses.

## 3. Results

### 3.1. The Characteristics of Study Population

A total of 495 doctors were enrolled in this study. The study population were widely distributed, covering 28 provinces and 78 cities in China (Figure 1A,B), with the majority in the southeast (47%), mainly from first-tier and second-tier cities (86%) (Figure 1C). The survey respondents mainly worked at the large medical centers (93%), of which the general hospitals were the majority (70%). Most participants were from breast surgery (46%) and medical oncology (39%) departments. Overall, 16% of the participants were professors, 28% of them were associate professors, and the proportion percentage of attending physicians was 43%.

### 3.2. Doctors’ Perceptions and Choices in China

Most clinicians (86%) agreed with the guidelines of Chinese Society of Clinical Oncology (CSCO) on HER-2 testing with the majority (91%) of hospitals having the ability to perform HER-2 testing. The primary influential factors for doctors in making treatment plans included the patients’ disease process, the performance status, and the economic status of patients, as shown in Figure 2.

Doctors valued OS and PFS most when they chose drugs for HER2-positive advanced breast cancer patients while the objective response rate (ORR), the QOL of the patients, the drug safety and the drug costs would also be taken into account (Figure 3). Although the guidelines did not recommend tyrosine kinase inhibitors (TKIs) for neoadjuvant or adjuvant anti-HER2 therapy, TKIs were used in clinical practice by 17% of doctors.

Regimen and treatment response to neoadjuvant or adjuvant chemotherapy were important factors in deciding the first-line treatment. For patients who received trastuzumab in neoadjuvant or adjuvant treatment and the disease-free interval (DFI) was less than 6 months, the most often used first-line treatment was single-agent chemotherapy plus trastuzumab and pertuzumab (59%), followed by single-agent chemotherapy plus TKI (48%). If the DFI was 6–12 months, 65% of doctors selected single-agent chemotherapy plus trastuzumab and pertuzumab, followed by single-agent chemotherapy plus TKI (54%). For patients with a DFI of more than 12 months, 77% of doctors selected single-agent chemotherapy plus trastuzumab and pertuzumab, followed by combined chemotherapy plus trastuzumab (59%). However, only 34% of respondents selected single-agent chemotherapy plus TKI. More details are shown in Figure 4. For patients with only bone metastases, the most commonly used first-line treatment regimens were docetaxel plus trastuzumab and pertuzumab (64%) and docetaxel plus capecitabine and trastuzumab (57%). Most doctors (67%) selected chemotherapy plus trastuzumab and pertuzumab as the first-line treatment while 43% and 32% of the participants would choose chemotherapy plus pyrotinib and chemotherapy with lapatinib, respectively, for patients with brain metastases. The main reasons for switching the original first-line treatment regimen were disease progression (83%) and economic factors (72%).

The majority of doctors believed that PFS of less than 6 months in the first-line setting indicated primary resistance to anti-HER2 treatment (71%) while 23% of doctors agreed that PFS of less than 3 months suggested primary resistance. For patients with PFS ≥ 6 months, 54% of doctors retained trastuzumab and replaced chemotherapy drugs in second-line treatment regimens. On the other hand, for patients with PFS < 6 months, most participants chose pyrotinib plus capecitabine as the second-line treatment regimen (52%). The detailed choices of respondents are shown in Figure 5.

Based on the above data, Chinese doctors made clinical decisions regarding the diagnosis and treatment of HER2-positive MBC following the current guidelines and general consensuses.

### 3.3. The Effect of the Economic Factors

Economic factors were attributed as being the primary reason for poor compliance of single-targeted and dual-targeted anti-HER2 treatment The patient’s disease process is the most important indicator for doctors when making treatment plans, but 37% of doctors would consider the financial status of patients. Physicians paid more attention to medical insurance than surgeons (72% vs. 64%). Overall, 53% of doctors considered the economic conditions of patients when they make decisions for the first-line treatment plan for HER2-positive MBC patients. Overall, 15% of doctors considered drug costs and 13% of doctors attached importance to the cost-effectiveness of the drugs.

### 3.4. Difference between First-Tier and Second-Tier or Other Cities

The proportions of patients tested for HER-2 status in either primary or recurrent/metastatic specimens from first-tier cities were 83% and 72%, respectively, which were significantly higher than those in the second-tier or other cities (73% and 61%, respectively). The proportion of patients receiving anti-HER2 targeted therapy in first-tier cities was higher than that in second-tier and other cities (73% vs. 67%). Doctors in second-tier or other cities paid more attention to the treatment expense for drug selection than those in first-tier cities (18% vs. 11%). Patients in first-tier cities received dual-targeted regimens longer than those in the second-tier or other cities (12 vs 9 months).

## 4. Discussion

This is the largest-scale survey conducted regarding the diagnosis and treatment of HER2-positive MBC patients in China. Most participants were attending physicians or those in senior positions whose medical skills reflected the local healthcare level. Moreover, the respondent hospitals were large medical centers covering 28 provinces across the country which had extensive national influence. Consequently, this survey was highly comprehensive.

Previous studies indicated that, for patients with stage IV breast cancer, the average expense allowed by insurance for each patient in the year after diagnosis was USD 134,682 [12]. Breast cancer is regarded as one of the costliest malignancies to treat, which puts a heavy financial burden on patients, their families, and healthcare systems all over the world [13,14,15,16,17]. The variation in the socioeconomic status of patients alone can result in different treatments and diverse prognoses [18]. Although health insurance in China has covered boarder regions and additional medication, there are still a number of anti-HER2 drugs failing entrance into the health insurance directory. For the HER2-positive MBC patients, treatment should be continuous and anti-HER2 drugs are indispensable to long-term therapy. However, several HER2-targeted drugs that are not reimbursed by medical insurance or even unavailable in China, are expensive for Chinese patients. China is still a developing country with disproportionation in economic development. Economic factors are bound to play a vital role in the treatment of HER2-positive MBC patients in our country. A cross-sectional study in China showed that the expenditure rate per patient with breast cancer was USD 8532 and even higher for MBC patients. In contrast, the average annual household income was only USD 8607 [19]. A recent survey on the financial hardship of Chinese cancer survivors reported that about 10% of patients have forgone treatment because of the cost [20]. Our research suggested that the economic conditions of patients did not allow for the continuation of medication as the main reason for poor compliance to the single-targeted and dual-targeted treatment. Overall, 37% of doctors would take into consideration the economic status of patients when making treatment plans and 53% of respondents took the patient’s financial situation into account during the decision-making for first-line treatment regimens. The common reason for changing the original first-line treatment regimen was economic factors (72%). When choosing the drugs for HER2-positive MBC patients, 15% of doctors would consider the medication expense while 13% of participants took the cost-effectiveness of the drugs into account. Taken together, the economic factors played an important role in the doctors’ clinical decisions making for patients with HER2-positive MBC which required expensive drugs.

We found the differences in treatment choices between respondents in the first-tier and the second-tier or other cities. Economic factors seemed to have a greater impact on the decision-making of doctors in the second-tier or other cities. Among the reasons for the discontinuation of the first-line treatment regimen was the higher proportion of doctors in the second-tier and other cities as economic factors. Doctors in the second-tier and other cities paid more attention to the treatment expenses when selecting therapeutic drugs in comparison to those in the first-tier cities. A lower proportion of patients in the second-tier or other cities could receive HER2-targeted drugs and patients in the second-tier or other cities accepted dual-target treatment for a significantly shorter time. As mentioned above, first-tier cities are in a leading position in the country economically and in health care. Therefore, we speculated that economic factors were the main reasons for the differences between the first-tier cities and the second-tier and other cities.

Encouragingly, some doctors would make clinical decisions based on the research evidence. TKIs were used for neoadjuvant or adjuvant anti-HER2 treatment by 17% of doctors. Several trials indicated that dual HER2 blockade with trastuzumab and lapatinib resulted in a significant improvement in the pathological complete response (pCR) rate and survival [21,22,23]. Pyrotinib-containing neoadjuvant therapy also showed favorable effectiveness with manageable toxicity [24,25]. However, there were no significant differences in disease-free survival (DFS) or OS between the dual-blockade group and the trastuzumab-alone group [26]. The ExteNET trial found benefits maintained in iDFS at 5 years (90.2% vs. 87.7%) for patients with HER2-positive breast cancer to receive extended adjuvant neratinib after a year of trastuzumab [27]. As such, TKIs may be considered for patients who do not respond well to the neoadjuvant treatment containing HER2 antibodies, patients who require short-term efficacy, or patients who have completed 1 year of trastuzumab in the adjuvant setting. However, our study did not investigate the specific clinical situations of TKIs selection further, hence the reasons why these doctors selected TKIs for the treatment of early-stage patients are unknown. For patients with bone-only metastases, the most commonly used first-line treatment regimen was chemotherapy plus HER2 antibodies. By contrast, for patients with brain metastases, 43% and 32% of the participants chose chemotherapy plus pyrotinib or lapatinib, respectively. It seemed that respondents tended to choose TKIs for patients with brain metastases. The brain is a sanctuary site for HER2-positive breast cancer [3]. Treatment options are still limited, and the prognosis remains poor for breast cancer patients with brain metastases. However, several studies indicated that anti-HER2 monoclonal antibodies and antibody-drug conjugates improved survival rates in patients with brain metastases, but their intracranial effects are controversial considering the large-molecule property that impedes the infiltration through blood–brain barrier [28,29,30]. From this perspective, TKIs may play an important role in the treatment of brain metastases. A real-world study has also shown the promising intracranial efficacy of pyrotinib. The ORR was as high as 66.7%, and three out of nine patients achieved complete remission of brain lesions when using the combination of pyrotinib-based systematic therapy with local treatment, which suggests this is an ideal treatment option of pyrotinib plus chemotherapy and radiotherapy for better intracranial control [31]. The PERMEATE study indicated that the intracranial objective response rate in patients with radiotherapy-naive HER2-positive brain metastases receiving pyrotinib plus capecitabine was 74.6%, and the intracranial objective response rate in patients who had progressive disease after radiotherapy was 42.1% [32]. Another real-world study included 15 lapatinib-treated patients with brain metastases and the median PFS of these 15 patients reached up to 6.0 months by treatment with pyrotinib [33]. The PHENIX study enrolling 31 patients with brain metastases indicated that the median PFS of the pyrotinib plus capecitabine group was 6.9 months [34], which was an improvement from that of lapatinib plus capecitabine group (5.5 months) and T-DM1 group in other studies [35,36]. Apart from the encouraging findings of the TBCRC022 [37], the Phase III NALA trial indicated that neratinib plus capecitabine resulted in a 24% reduction in the risk of disease progression and a trend toward improved OS when comparing to lapatinib plus capecitabine [38]. Moreover, time for intervention for symptomatic central nervous system disease was also delayed [39]. The HER2CLIMB study indicated that tucatinib plus trastuzumab with capecitabine produced improved PFS and OS than trastuzumab with capecitabine in pretreated HER2-positive MBC patients with brain metastases [40]. The Phase II study (LANDSCAPE) showed that 65.9% of MBC patients, including heavily pretreated patients, achieved a more than 50% volumetric reduction in brain lesions treated with lapatinib and capecitabine, even without prior radiotherapy [35]. Furthermore, the improvement of the median OS in patients treated with lapatinib and capecitabine was revealed, in comparison to patients that retained trastuzumab-based therapies (27.9 vs 16.7 months) [41]. In conclusion, this evidence suggests that TKIs have particular advantages in the treatment of HER2-positive breast cancer patients with brain metastases.

For patients who have received trastuzumab in neoadjuvant or adjuvant treatment and regardless of how long the DFI was, most doctors chose single-agent chemotherapy plus trastuzumab and pertuzumab as the first-line treatment regimen, which was not consistent with the guidelines. The deviation might be due to the following reasons: this was a multiple-choice question and respondents could choose five kinds of regimens. T-DM1 was still unavailable in China when conducting this investigation while trastuzumab and pertuzumab were both available and commonly used as HER2-targeted drugs. More doctors chose TKI-containing regimens according to the shorter DFI. For patients with a DFI of more than 12 months, 34% of participants selected single-agent chemotherapy plus TKI while for patients with DFI of less than 12 months, over 48% of respondents tended to choose single-agent chemotherapy plus TKI.

Breast cancer is a heterogeneous disease composed of a growing number of recognized biological subtypes. The prognostic and etiologic importance of this diversity is complicated by many factors, including the observation that differences in clinical outcomes often correlate with race. Data from Surveillance, Epidemiology, and End Results (SEER) registries in 2010 suggested that there were differences in the expression of HER2 status among different races [42]. HER2-positive / HR-negative patients were more likely to be non-Hispanic black (OR = 1.4, 95% confidence interval [CI] = 1.2 to 1.6), non-Hispanic Asian Pacific Islander (OR = 1.8, 95% CI = 1.5 to 2.1), and Hispanic (OR = 1.4, 95% CI = 1.2 to 1.6) rather than non-Hispanic white (referent). However, the etiologic basis for different racial and ethnic patterns remains unclear. Looking carefully at individual risk factors, such as reproductive history, lactation, weight, physical activity, mammography, postmenopausal hormone use, and longevity, could explain the apparent differences in the diagnosis of breast cancer subtypes by race and ethnicity in SEER areas [43]. In addition, Surbhi Bansil et al. found a significantly greater amount of tumor-infiltrating lymphocytes (TILs) in Asians (37.7%, *p* = 0.01) and Native Hawaiian/Pacific Islander (37.2%, *p* = 0.02) patients compared to White patients [44]. However, the percentage of TILs in breast cancer tumors of all patients based on race and breast cancer subtype showed that there were no significant differences in the expression of TILs found among all races with HER2-positive breast cancer. Unfortunately, our study did not include the investigation of differences in HER2-positive breast cancer within Asian populations. Further studies are required to explore the inherent biological characteristics of HER2-positive breast cancer in Asian populations to derive relevant implications.

One limitation of this survey was responder bias. The participants of this study were mainly from major hospitals in large cities and the diagnosis and treatment situation in the second-level hospitals may differ in certain areas. Additionally, the observational and retrospective design of the study provided weak evidence in terms of causality and was susceptible to memory bias. Nevertheless, the present study had certain merits. To the best of our knowledge, this is the first survey to focus on the situation surrounding the diagnosis and treatment of HER2-positive metastatic breast cancer patients in China. Moreover, a total of 495 doctors, most of whom were attending physicians or of a higher position, were enrolled in this study and the respondent 203 hospitals were large in capacity and spanned 28 provinces across the country. Therefore, our study is highly comprehensive as it covers a broad geographic range and investigates a relatively large number of experienced doctors and large medical centers, which makes the results more compelling.

## 5. Conclusions

We investigated nearly 500 doctors from 203 centers covering 78 cities across 28 provinces in China. Our study provided comprehensive insight into the current situation regarding the diagnosis and treatment of HER2-positive MBC patients in China. The results showed that Chinese doctors’ decisions were generally in accordance with the guidelines and overall consensuses; however, their clinical decisions were constrained by real-world factors, such as treatment expenses, medical insurance, and drug accessibility. Doctors have to consider economic factors in the management of HER2-positive MBC patients. The encouraging point is that some doctors make the treatment decisions based on the latest evidence. This survey’s results will provide a reference for consensuses and guidelines on the diagnosis, treatment standards, and the development of future specialized training programs for HER2-positive MBC. Further research encompassing more intensive investigations and covering a broader geographic range is required to show the overall landscape for diagnosis and treatment of HER2-positive MBC practiced in China.

## Figures and Tables

**Figure 1 jpm-13-00365-f001:**
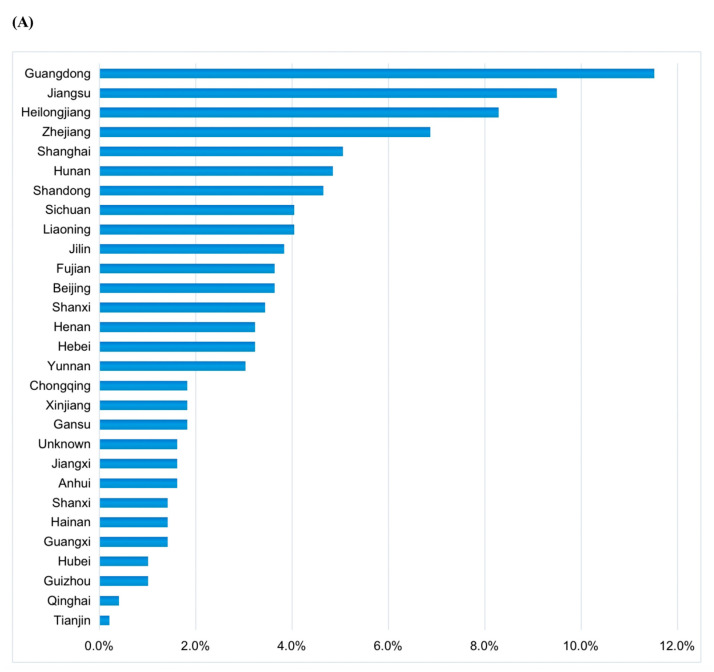
(**A**) The respondent hospitals covered 28 provinces in China; (**B**) number of hospitals sampled from each province; and (**C**) the level of the cities involved in this study.

**Figure 2 jpm-13-00365-f002:**
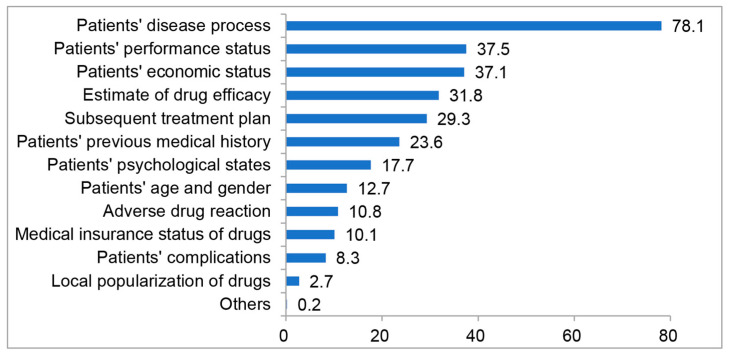
The average score based on the considerations of 495 doctors making treatment plans for HER2-positive metastatic breast cancer patients. (It is a ranking question requiring selection of 5 options, which are rated as 100, 80, 60, 40, 20 scores and 0 score for those are not selected).

**Figure 3 jpm-13-00365-f003:**
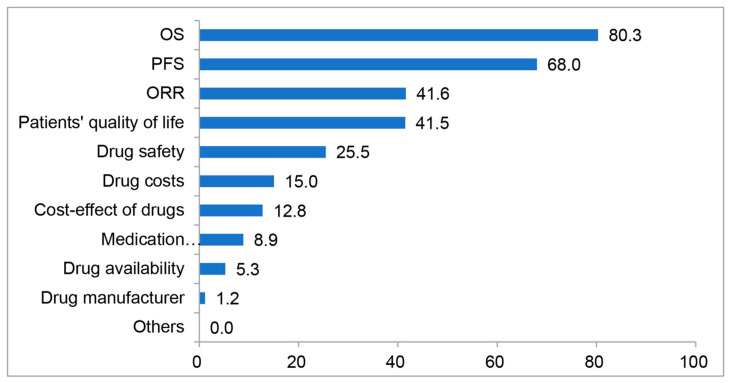
The average score based on the considerations of 495 doctors when choosing drugs for HER2-positive metastatic breast cancer patients. (It is a ranking question requiring selection of 5 options, which are rated as 100, 80, 60, 40, 20 scores and 0 score for those are not selected Doctors selected up to 5 items and scored 100, 80, 60, 40, and 20, respectively).

**Figure 4 jpm-13-00365-f004:**
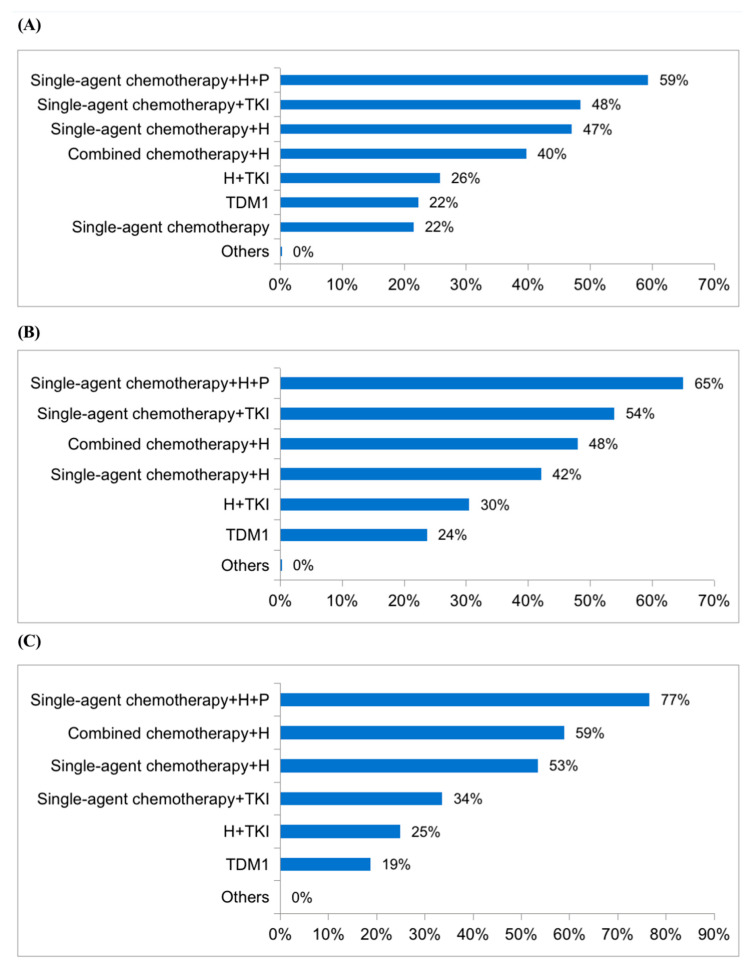
The first-line treatment regimens chosen by 423 doctors for HER2-positive MBC patients who received trastuzumab in neoadjuvant or adjuvant treatment: (**A**) DFI < 6 months; (**B**) 6 months ≤ DFI ≤ 12 months; (**C**) DFI > 12 months.

**Figure 5 jpm-13-00365-f005:**
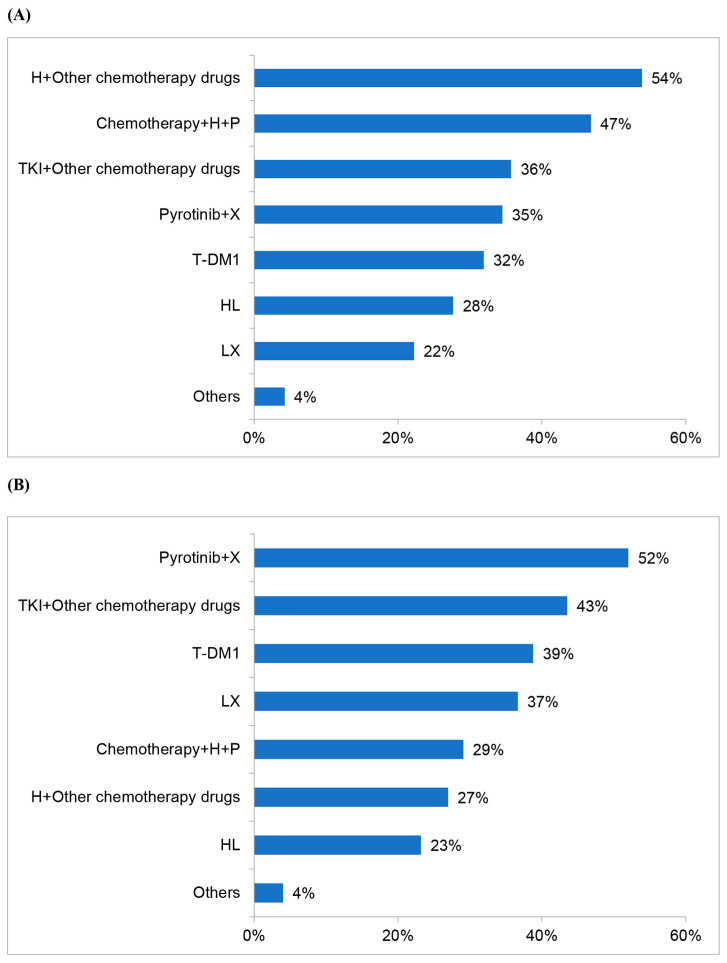
The second-line treatment regimens chosen by 423 doctors for HER2-positive MBC patients: (**A**) PFS in the first-line setting ≥ 6 months; and (**B**) PFS in the first-line setting < 6 months.

## Data Availability

The data used and/or analyzed during the current study are available from the corresponding author on reasonable request.

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
