# Peer review of "Current Situation of Diagnosis and Treatment of HER2-Positive Metastatic Breast Cancer Patients in China: A Nationwide Cross-Sectional Survey of Doctors"

_jpm, 2023, doi:10.3390/jpm13020365_

Round 1
Reviewer 1 Report
The manuscript highlights the choice of treatment for patients with metastatic breast cancer. A huge cohort of doctors (almost 500 specialists), mostly from large hospitals in the first or second-tier Chinese cities, answered the questionnaires which were focused on the main factors impacting the doctors’ clinical decision to select the most efficient treatment scheme.
While the methodology of the study is interesting to physicians in different countries, the obtained results are relevant mostly for the Chinese doctors. The results can be easily foreseen: the economic factors directly depend on the size and status of the city and are obviously connected with the welfare related to the wellbeing of the citizens.
Since the manuscript is to a great degree a statistical work, the references should contain the most recent data. Still many cited works are dated earlier than 5 years ago and need an improvement.
Despite this fact the study is well designed and performed, and the manuscript may be published in a medical journal of anticancer profile.
Author Response
We really appreciate the editor and the reviewers for their constructive comments and suggestions on our manuscript. We have revised the manuscript according to the reviewer’ s comments and cited additional literature within 5 years. The changes to our manuscript was highlighted in the revised manuscript by using colored text in revisions mode.
Reviewer 2 Report
This article presents a survey on the Current Situation of Diagnosis and Treatment of HER2-positive Metastatic Breast Cancer Patients. Breast cancer is widespread and has become a global issue that causes hundreds of women's deaths each year.
1) (a) What is the main question addressed by the research? Basically, the author addresses one of the most alarming cancer i.e. breast cancer. In China that has been increasing in recent years, with about 279,000 new cases each year. Approximately 20-30% of advanced breast cancer patients were Human epidermal growth factor receptor 2 (HER2)-positive. The main question addressed by the author is the diagnosis and treatment of HER2-positive metastatic breast cancer (MBC) patients. (b). Is it relevant and interesting? The topic is really very interesting, because of The electronic questionnaires, including the basic information of respondents, characteristics of patients, and the present status of diagnosis and treatment, were sent to 495 doctors of 203 medical centers covering 28 provinces in 2019, Author also share the results of that questionaries by covering treatment plans included the disease process, the performance status, and the economic status of patientsand it has been found that Economic factors seemed to have a greater impact on the decision-making of doctors. (c). How original is the topic? I personally believe the topic is original and new in medical world. Very limited work has been done in literature specially survey on such novel topic. (d). What does it add to the subject area compared with other published material? Other publish work is focusing on other treatments and cancer types like: lapatinib and capecitabin, positive metastatic breast cancer and central nervous system metastases, Promising Efficacy in Lapatinib-Treated Patients and in Brain Metastasis and metastatic breast cancer treatment with trastuzumab and taxanes. while this research paper only focus on Diagnosis and Treatment of HER2-positive Metastatic Breast Cancer Patients. Author also compares the economic factor too which is missing in mostly in literature. (e) Is the paper well written? I personally believe that Paper is well written and organized. (f) Is the text clear and easy to read? Yes. (g) Are the conclusions consistent with the evidence and arguments presented? The author supports the research (evidence and arguments presented) with pie and bar chart results and electronic questioners. Which depict the deep work and literature done by the authors. (e). Do they address the main question posed? Yes, the main question is addressed as the results showed 289 that the choices of Chinese doctors were basically in accordance with the guidelines and 290 consensuses, but their clinical decisions making was also constrained by realistic factors 291 such as treatment expense, medical insurance, and drug accessibility. Doctors have to consider economic factors in the management of HER2-positive MBC patients. Personal Note: I personally agree that the authors have address all the important point and up to best of my knowledge I answer all the questions.
However, it is recommended that a few more references may be included in the introduction part. For example:
[1] Abdul Halim, Ahmad Ashraf, et al. "Existing and Emerging Breast Cancer DetectionTechnologies and Its Challenges: A Review." Applied Sciences 11.22 (2021): 10753.
Author Response
Thank the reviewer very much for the positive comments on our manuscript. We have included the reference entitled ”Existing and Emerging Breast Cancer Detection Technologies and Its Challenges: A Review.” as Reference 4 in the introduction part. The changes was highlighted in the revised manuscript.
Reviewer 3 Report
I read with great interest the manuscript entitled "Current Situation of Diagnosis and Treatment of HER2-positive Metastatic Breast Cancer Patients in China: A Nationwide 3 Cross-Sectional Survey of Doctors”. In this study, the author investigated nearly 500 doctors from 203 centers covering 78 cities from provinces in China. The study provided comprehensive insight into the current situation of diagnosing and treating HER2-positive MBC patients in China. The results showed that Chinese doctors’ choices followed the guidelines and consensuses. Still, their clinical decisions making was also constrained by realistic factors such as treatment expense, medical insurance, and drug accessibility. The manuscript is well written, and it is a very current topic. Still, I have some concerns about improving the quality of the manuscript. I have mentioned my comments in the manuscript file; if authors give attention to those questions should be welcomed.

Author Response
Thank you so much for your constructive comments and suggestions on our manuscript. We have modified the Figure 1 and the manuscript as you suggested. You may see more details in the revised manuscript. We thank the editor and reviewers again for the constructive suggestions that have improved both the quality and the clarity of our work, and we hope that the new version could be up to your standard.
Round 2
Reviewer 3 Report
The authors made significant revisions based on the reviewer’s comments. The manuscript can be accepted for publication.